# Cd Phytoextraction Potential in Halophyte *Salicornia fruticosa*: Salinity Impact

**DOI:** 10.3390/plants11192556

**Published:** 2022-09-28

**Authors:** Fawzy Mahmoud Salama, Arwa Abdulkreem AL-Huqail, Mohammed Ali, Amany H. A. Abeed

**Affiliations:** 1Department of Botany and Microbiology, Faculty of Science, Assiut University, Assiut 71516, Egypt; 2Department of Biology, College of Science, Princess Nourah bint Abdulrahman University, P.O. Box 84428, Riyadh 11671, Saudi Arabia; 3Egyptian Deserts Gene Bank, North Sinai Research Station, Department of Genetic Resources, Desert Research Center, Cairo 11753, Egypt

**Keywords:** antioxidants, halophyte, phytochelatins, *Salicornia fruticosa*

## Abstract

The phytoextraction potential of halophytes has been broadly recognized. Nevertheless, the impact of salt on the accumulation proprieties of cadmium (Cd) in different halophytic species, likely linked to their salt tolerance, remains unclear. A hydroponic culture was used to investigate the impact of salinity on Cd tolerance as well as accumulation in the distinct halophyte *Salicornia fruticosa* (*S. fruticosa*). The plant was subjected to 0, 25, and 50 μg L^−1^ Cd (0-Cd, L-Cd, and H-Cd, respectively), with or without 50, 100, and 200 mM NaCl in the nutrient solution. Data demonstrated that Cd individually induced depletion in biomass accumulation. NaCl amplified the Cd tolerance induced by enhanced biomass gaining and root length, which was associated with adequate transpiration, leaf succulence, elevated levels of ascorbic acid (ASA), reduced glutathione (GSH), phytochelatins (PCs), and proline as well as antioxidant enzymatic capacity via upregulation of peroxidases (PO), glutathione peroxidase, ascorbate peroxidase, and superoxide dismutase. All Cd treatments decreased the uptake of calcium (Ca) as well as potassium (K) and transport to the shoots; however, sodium (Na) accumulation in the shoots was not influenced by Cd. Consequently, *S. fruticosa* retained its halophytic properties. Based on the low transfer efficiency and high enrichment coefficient at 0–50 mM NaCl, an examination of Cd accumulation characteristics revealed that phytostabilization was the selected phytoremediation strategy. At 100–200 mM NaCl, the high aboveground Cd-translocation and high absorption efficiency encourage phytoremediation via phytoextraction. The results revealed that *S. fruticosa* might be also potentially utilized to renovate saline soils tainted with heavy metals (HMs) because of its maximized capacity for Cd tolerance magnified by NaCl. Cd accumulation in *S. fruticosa* is mainly depending on the NaCl concentration. Future studies may be established for other heavy metal pollutants screening, to detect which could be extracted and/or stabilized by the *S. fruticosa* plant; moreover, other substrates presenting high electrical conductivity should be identified for reclamation.

## 1. Introduction

Cadmium (Cd) is one of the most potentially toxic elements, which seriously increases environmental pollution and affects plant life. Cd is identified as a carcinogenic inducer; it ranks 7th among the top 10 toxicants [1]. It extensively bioaccumulates in soils and water bodies from inappropriate anthropogenic routes such as fertilizers, pesticide impurities, and the extensive use of ash and irrigation with waste water [2,3]. Cd is not degradable or modifiable; thus, it is greatly persistent and accumulates in the environment [2]. Owing to the high solubility and mobility of Cd in soil sediments, it is easily taken up by plants and then translocates to the aerials and, hence, enters the food chain. Thus, it is vital to restrict Cd toxicity to guarantee sustainable plant production and, because of that, the United States Environmental Protection Agency (USEPA) has considered the concentration of 3 mg Cd kg^−1^ as the maximum allowed level in agricultural soils receiving sludge [4].

Unfortunately, most of the Egyptian coastal zones along the Mediterranean Sea, recognized as marine habitats, are subjected to intense discharges of pollutants from numerous anthropogenic activities [5]. Following industrialization, abnormal quantities of heavy metals (HMs), recently termed as potentially toxic elements [6], such as Cd, Cu, Pb, Ni, and Zn have been released and continue to be released into the aquatic environment through storm water and wastewater discharges. The main sources of Cd pollution come from batteries and electrical sources; pigments and paints; alloys and solids; fuels; plastics; and fertilizers [7]. El-Sorogy et al. [8] have documented that the Cd levels (amounted as 19–44 µg L^−1^) recorded in some coastal zones along the Mediterranean Sea are higher than the permissible limits recommended by the USEPA [9] and Egyptian laws [10]. Accordingly, such coastal areas are simultaneously affected by salinity and heavy metal deposition. This causes the saline coastal areas polluted with potentially toxic elements to arise as a global environmental problem [11]. 

Researchers and government agencies have recently paid particular attention to the elevated Cd concentration in the environment as a concerning issue. The current physical and chemical practices of removing Cd contamination from the plant habitats are complicated processes with high costs [12]. Alternatively, phytoremediation technology is one of the most widely used techniques in this regard. Due to the inherent potential of some plant species to collect certain toxic elements such as Cd from their surroundings, the interest in utilizing such plants for substrates rehabilitation has lately been expanded. Phytoremediator species are valued for their relatively low cost and high safety [13]. The ideal candidates for phytoremediation are plants characterized by high productivity and efficiency in transferring heavy metals from cultivated soil to the epigeal biomass [14]. Halophytes are characterized as plants that can survive and reproduce in environments ranging from normal to severely saline. Monocotyledonous halophytes display optimum growth in 50–100 mM of NaCl, whereas dicothalophytes are more tolerant as compared to monocot species and show optimum growth in 100–200 mM NaCl [14]. Salt amendment accomplishes the salt requirement for maximum growth, thus maintaining high-yields of aboveground biomass for HM deposition in this part that is further harvested [15]. 

Interestingly, the wetland species of halophytes have proven their capability to accumulate significant levels of toxic elements in their tissues [15] and remediate HMs from salinized soil and water as well [16]. The high-tolerance aspects by halophytes of metals most likely correlate with a progressed salt-tolerance mechanism, such as antioxidant systems [17]. Moreover, osmo-protectants production, such as proline to scavenge free radicals as well as retrain the balance of water [17,18] and salt gland excretion onto the surface of the leaf, thus, plays a crucial role in tolerance induction. In addition to NaCl, this mechanism enables the extraction of inorganic contaminants [18]. Several halophytes showed improved Cd accumulation and translocation from the roots to the shoots under NaCl conditions [14]. Numerous explorations revealed that adding salt (NaCl) to the medium increased Cd phytoextraction [19]. For example, Sepehr et al. [20] illustrated that salinity alleviated Cd toxicity in maize plants. It seems that moderate NaCl levels are hypothesized to enhance plants’ growth, besides protecting against HM toxicity via modulating osmotic adjustment and ion uptake and stimulating antioxidant mechanisms [21]. Therefore, halophytes are the most desired plant species for remediating HMs such as Cd-contaminated salty media.

*Salicornia fruticosa*, termed glasswort, is an annual succulent euhalophyte belonging to the Chenopodiaceae family. As a wetland plant, it grows on muddy seashores as well as in saline marshes [22]. Although the plant exhibits a great tendency to grow well across various concentrations of soil salt up to 8% [23], its HMs-remediation capability is still not well-investigated, so there is a paucity of data on the metabolic responses to the combined stress of Cd and NaCl. In addition, there is currently no evidence of the response to Cd absorption, translocation, and, therefore, phytoremediation under NaCl stress. Hence, the current research could provide a fundamental basis of interpreting the NaCl impact on Cd uptake, accumulation, and translocation in recently screened *S. fruticosa*. Furthermore, Cd and NaCl metabolic responses, as well as the connection between Cd and metabolites in plants under Cd stress, were studied in the absence and presence of NaCl, suggesting the possibility of using *S. fruticosa* as a Cd phytoremediator.

## 2. Results

### 2.1. Phenotypic Criteria Affected by Interaction between Cd Stress and Salinity in S. fruticosa

Morphologically, plants treated with L-Cd showed initial chlorosis that worsened when raising the concentration of Cd, before they developed necrosis with abscission as well as leaf senescence. In the absence of Cd, 100 and 200 mM NaCl did not impact *S. fruticosa* morphology. The difference was remarkable, particularly between high-salt-treated and low-salt-treated plants receiving 50 μg L^−1^ of Cd. In the absence of Cd, all plants showed high growth tendency and enhanced phenotypic criteria, in terms of aboveground biomass and root length, throughout the experimental period for up to 200 mM NaCl (Figure 1a,b). Treatment, with H-Cd but without NaCl, negatively impacts aboveground biomass and root length; however, L-Cd did not affect biomass and slightly stimulated root length. The addition of NaCl substantially enhanced root length and the development of plant biomass and restored adequate plant growth; however, plant growth response to the Cd and NaCl combination depends on the concentration of NaCl. High-salt-treated plants had better growth than low-salt-treated plants.

### 2.2. Water Relation Indices Affected by Cd Stress and Salinity Co-Occurrence in S. fruticosa

The transpiration rate of plants treated with different salt concentrations remained unaffected, compared with NaCl-untreated plants (Table 1). Alone, L-Cd treatment exhibited a slight reduction in *S. fruticosa* transpiration rate, while H-Cd severely inhibited this trait. Elevating salt concentration in the medium from 50 to 200 mM significantly restored the transpiration rate of the H-Cd-treated plant, whereas no substantial effect was detected in plants grown in L-Cd. Shoots from different salt-concentration-treated plants did not differ in succulence (Table 1). Alone, L-Cd treatment displayed an unchanged succulence degree for the shoots in comparison with the leaves of 0-Cd + NaCl-untreated plants, whereas H-Cd severely reduced this trait (Table 1). NaCl co-occurrence markedly restored shoot-succulence degree. The TOP value was sustained for plants grown along all NaCl concentrations (Table 1). Increasing doses of Cd alone may induce consequent osmotic stress that elicits the importance of increasing the TOP. Co-occurrence of salt effectively reduced the TOP value to that of the corresponding salt-treated plant.

### 2.3. Mineral Composition Affected by Cd Stress and Salinity Co-Occurrence in S. fruticosa

All treatments of Cd alone reduced Na, K, and Ca shoot concentrations (Figure 2a–c). A progressive decrement was detected in Ca and K accumulation, in accordance with the higher Na accumulation as the external NaCl supply increased. However, increasing the amount of Cd from 25 to 50 μg L^−1^ did not alter the accumulation of Na in the shoots. The simultaneously imposed Cd and NaCl, compared to those submitted to Cd alone, reduced Ca and K concentrations in the shoots further. In non-saline conditions, the concentrations of Cd in roots and shoots elevated with the elevation in Cd supply and were substantially elevated in the roots compared to the shoots (Figure 2d). Compared to Cd alone, the addition of NaCl substantially enhanced the Cd concentration. Plants of *S. fruticosa* grown in saline media acquired higher amounts of Cd compared with plants grown in non-saline media (Figure 2d). Root Cd concentration was not affected by NaCl with H-Cd treatment. L-Cd moderately increased the Cd concentration in roots with NaCl; however, elevated NaCl concentrations did not change root Cd concentration.

### 2.4. Phytoremediation Parameters Affected by Cd Stress and Salinity Co-Occurrence in S. fruticosa

In non-saline conditions, the bioaccumulation factor (BCF) and translocation factor (TF) values were unaffected (Table 2). For both Cd treatments (25 or 50 μg L^−1^), increasing the salt content in the medium from 100 to 200 mM increased the Cd translocated in the shoots and decreased the Cd retained in the roots (Table 2), but, at 50 mM NaCl, the majority of Cd was allocated in the roots rather than the shoots. Increasing the salt concentration substantially improved the amounts of Cd deposited in the shoots. This increase was essentially because of the elevated biomass production in the plants subjected to the combined effect of NaCl and Cd. Furthermore, as evidenced by the elevation in TFs as well as BCFs, NaCl treatment substantially enhanced Cd absorption and translocation. TF was greater in plants receiving the Cd and NaCl mixture than in those receiving only Cd (Table 2). Elevating the concentration of salt in the medium from 100 to 200 mM resulted in more Cd transported from the roots to the shoots. Therefore, factors of translocation were highest in plants receiving 200 NaCl. The Cd absorption efficiency (AE) of this halophyte was measured further to assess the potential and efficacy of root Cd absorption. Under non-saline conditions, the AE of *S. fruticosa* was substantially improved when raising the Cd stress (*p* < 0.05). NaCl application further elevated the AE of *S. fruticosa* in L-Cd and H-Cd combined with 200 mM, compared with that in L-Cd and H-Cd without salt.

### 2.5. Non-Enzymatic Antioxidant Indices as Affected by Cd Stress and Salinity Co-Occurrence in S. fruticosa

Plants only treated with L-Cd had a higher content of low molecular weight antioxidant ascorbic acid (ASA), whereas this trait was depleted by H-Cd (Table 3). NaCl imposition demonstrated no change in ASA content, regardless of the dose of NaCl. ASA was strongly increased by salinity and L-Cd co-occurrence, whereas it decreased again to values comparable to 0-Cd + NaCl-treated plants, with combined H-Cd and NaCl. 

All the study treatments significantly enhanced the reduced glutathione GSH content (Table 3). The highest GSH content was recorded for combined NaCl and Cd treatments, followed by that of the 0-Cd + NaCl-untreated plants, whereas the lowest GSH increment was recorded for NaCl-treated plants without a Cd supply. In contrast to GSH (despite their co-regulation), phytochelatins (PCs) showed a slight reduction under salinity and Cd co-occurrence. PCs were triggered due to Cd treatment alone, whereas no substantial change was recorded for 0-Cd + NaCl-treated plants (Table 3).

Cd treatment triggered proline accumulation. This response was Cd-dependent, with H-Cd inducing more proline than L-Cd (Table 3). However, all NaCl-treated plants exhibited a slightly nonsignificant increase in proline accumulation, compared to NaCl-untreated plants. NaCl occurrence efficiently ameliorated Cd impact on proline accumulation, while NaCl- and Cd-treated plants had a reduced proline content compared to plants treated only with Cd.

### 2.6. Alternations in the Capacities of Enzymatic Antioxidant of S. fruticosa as Affected by Cd Stress and Salinity Co-Occurrence

The activities of superoxide dismutase (SOD), glutathione peroxidase (GPX), ascorbate peroxidase (APX), and peroxidases (POs), along with an elevation in external Cd concentration (Figure 3a–c), were substantially improved. However, elevated supplementation of NaCl showed no change in their pattern, compared to NaCl-untreated plants, during the 17-day period of experimentation. Further increase in the enzymes’ activity was recorded for combined salinity and Cd-stressed plants. In contrast to SOD, APX, and GPX, peroxidase (PO) activity was reduced significantly by NaCl and Cd co-occurrence, in accordance with the highest PO activity for treatments of Cd alone, and minimized PO activity in salinized plants, for the former (Figure 3d).

## 3. Discussion

### 3.1. Phenotypic Criteria Affected by Interaction between Cd Stress and Salinity in S. fruticosa

To our knowledge, this is the first time to screen *S. fruticosa* for its Cd phytoremediation potential under the NaCl effect and to shed light on its operation of several basic biochemical tolerance mechanisms that may provide an advantage to this halophyte, with respect to heavy metals as co-environmental factors. In the current investigation, the large aboveground canopy and deep root system of *S. fruticosa* qualified it for Cd phytoremediation in polluted soil. According to Eissa and Abeed [2], plants with strong, deep roots may be employed in the phytoremediation of contaminated sites. Under our investigation, the data showed that Cd had toxic effects on aboveground biomass and root length, but the addition of NaCl decreased the negative symptoms of Cd treatment, and this modified effect increased with increasing NaCl concentration. Previous reports illustrated the contribution role of salinity in modulating the deleterious responses to Cd [24]. In the present study, *S. fruticosa* showed optimum growth when it was exposed to 200 mM NaCl, exhibiting higher salt tolerance and being able to yield a high amount of extraction parts (plant tissues for harvest), even for the H-Cd. Similarly, Ghnaya et al. [15] found that salinity, especially that which was optimal for growth, clearly improved the growth of *Sesuvium portulacastrum* under Cd stress.

### 3.2. Water Relation Indices Affected by Cd Stress and Salinity Co-Occurrence in S. fruticosa

Transpiration is a significant factor of the ion transport necessary for salt tolerance, since it allows xylem ions to move from the root cells to the stem cells. Inhibition in transpiration in plants treated with H-Cd alone influences the mobility and accumulation of Cd in shoots; thus, Cd was retained in the roots. In consequence, high Cd accumulation in the roots may cause Cd toxicity that inhibits the root apex growth and causes water uptake dysfunction, which is evident by a low water content and a reduced root system [25]. Increases in transpiration and mass flow, as well as photosynthesis, provide energy and oxygen for the active mobilization of salt. Sruthi et al. [26] suggested that heavy metals may have higher mobility under saline conditions due to an increase in transpiration, leading to a higher flux of metals into the plant.

NaCl-treated *S. fruticosa* plants exhibited degrees for shoot succulence. This capacity to retain water content showed that salt tolerance in this species is partially attributable to its capacity to accomplish an osmotic adjustment. Succulence tends to lower transpiration requirements by decreasing leaf heating [27]. Salinity considerably elevated the shoot succulence degree of *S. fruticosa*, which maintains its halophytic character even in the presence of Cd. In fact, plants cultivated under 100 and 200 mM NaCl exhibited higher shoot succulence degrees than plants grown in 50 mM NaCl, which may be explained by the substantial Na and K accumulation in the leaves of plants treated with high elevated salt, which was around double that of plants treated with a lower salt concentration.

The sustained total osmotic potential (TOP) value of plants grown along all NaCl concentrations may indicate that these plants are suffering less from osmotic stress, as they are grown in preferable salt concentrations; moreover, the high recorded succulence degrees resulted in cell sap dilution, thereby causing an adequate TOP value. Increasing doses of Cd alone may induce a consequential osmotic stress that elicits the importance of increasing the total osmotic potential. Co-occurrence of salt, effectively the TOP value, thus, may direct the whole cell energy towards normal plant growth, rather than manufacturing the costive osmolytes that participated in the induced total osmotic potential; this was advocated by the high Cd-induced proline in plants stressed by H-Cd alone. The alternations in the TOP value, due to the presence or absence of Cd in saline soil, indicated the high adaptive potentiality of *S. fruticosa*, and NaCl is shown to play a key role in the modulation of plant responses to Cd. Hamed et al. [28] suggested that the indirect contribution of Na to osmotic adjustment is a possible mechanism of resistance by *Sesuvium portulacastrum* to the combination of salinity and Cd.

### 3.3. Mineral Composition Affected by Cd Stress and Salinity Co-Occurrence in S. fruticosa

All halophytes must meet the challenge of osmotic adjustment to water stress by the high contribution of cations and anions (Na, K, and Ca contributed 67% of the solute concentration (molar in the shoot water) [16]. Diminishment in these inorganic fractions causes a failure of the efficient cellular osmotic adjustment, which was shown by the effect of Cd stress evident with the high total osmotic potential. Leaves have been shown to accumulate high amounts of Na and Cl, compartmentalizing these ions to vacuole lessens the osmotic potential of cells under saline conditions. Thus, Na rather than K was the ion involved in leaf succulence, shoot development, and cell expansion. Under 0-Cd + NaCl-treated plants, Na uptake was antagonistic to K uptake. It has also been reported that K uptake is adversely affected by NaCl exposure in *Atriplex portulacoides* [29]. In accordance with this, reductions in the K and Ca levels were observed in our plants at all salinity levels. It seems, therefore, that the halophytic proprieties of *S. fruticosa* are preserved in the presence of salt and Cd, and this was emphasized by the reduced K concentrations.

### 3.4. Phytoremediation Parameters Affected by Cd Stress and Salinity Co-Occurrence in S. fruticosa

Cd is mainly concentrated in the roots rather than in the shoots, indicating that the root system is the primary organ for Cd storage in *S. fruticosa*. The current study has shown that, in addition to alleviating the effects of Cd on plant growth, NaCl modified the uptake of Cd and its transport from the roots to the shoots. Notably, higher NaCl had no impact on Cd concentrations in the roots of *S. fruticosa*. We may, thus, deduce that the *S. fruticosa* plant is likely to transfer greater amounts of Cd to the shoots under saline conditions, because both higher water content and biomass possibly allow this species to tolerate and reserve more Cd in the shoots. The total amount of Cd accumulated in the shoots is the most important parameter to evaluate the potential of Cd extraction capacity in plants, representing the product of the shoot biomass by their Cd concentration. The ability of plants to translocate Cd from the roots toward the shoots was evaluated using the translocation factor, calculated as the ratio between Cd quantities in the shoots and roots. Higher TFs and shoot Cd accumulation (measured per plant) also suggest that greater Cd was shifted to the shoots in *S. fruticosa*. This may be associated with the fact that increasing NaCl provides more free Cl ions, which are available in the medium to mostly form Cd-Cl complexes. Cd-Cl complexes are mostly considered phytoavailable and are easily uptaken by plants, contributing to increased Cd absorption. At low salt concentrations, the intensity of Cd stress tolerance is very limited for a large extent; thus, immobilization of HM in the root system was observed, which may be considered a strategy for counteracting HM toxicity in photosynthesizing organs [30,31], so the adopted phytoremediation strategy could be accomplished through phytostabilization.

### 3.5. Non-Enzymatic Antioxidant Indices as Affected by Cd Stress and Salinity Co-Occurrence in S. fruticosa

Proline have been associated with the capacity of halophytes to tolerate salt stress by acting as an intracellular osmotic solute. In the current study, however, nonsignificant proline accumulation in response to salinity indicates that *S. fruticosa* has been able to tackle salinity without expending energy or any damage to the plant’s organs [28]. Similar results were cited by Samiei et al. [32] for *Climacoptera crassa*. Moreover, Parida and Jha [33] reported that 200 mM NaCl did not induce an increase in proline in *Salicornia brachiata* and that proline produced under high salt treatments (400 mM) possibly plays a more important role in protecting the enzymatic system in the cytoplast but not in adjusting the osmotic homeostasis. Previous studies have shown a high correlation between the intensity of Cd stress in the plant and the amount of proline production in the halophytes, including for *Juncus gerardi*, *Sesuvium portulacastrum*, and *Climacoptera crassa* [32,34,35]. Therefore, the present study shows that the increase in proline content is one of the main mechanisms of *S. fruticosa* to deal mostly with heavy metal stress rather than salinity, while heavy metal stress induced higher proline production compared to salinity. Clemens [36] suggested that while heavy metals evoked proline accumulation in plants, this is not directly originated from heavy metals stress, as water balance disturbance is responsible for the accumulation of proline. Water stress mediated by HMs necessitates proline production and biosynthesis, which are energy costive processes at the expense of cell growth. In our study, however, salinity co-occurrence efficiently serves as an energy saver by reducing proline level. This decline could be attributed to the enhanced exploitation of carbon skeletons to sustain growth in a toxic environment. We can deduce that proline, herein, acts as an osmoticum rather a ROS scavenger, since similar results were cited by Wiszniewska et al. [31]. In contrast, Lefèvre et al. [11] found higher proline concentrations in the leaves of Cd + NaCl-treated *Atriplex halimus* plants than in those of NaCl-treated ones.

Glutathione, ascorbate, and essential non-enzymatic antioxidants are also vital molecules to scavenge ROS, which cannot be detoxified by the enzymatic system. In the present study, salinity reduced the oxidative stress in *S. fruticosa* by enhancing glutathione and glutathione reductase activity, in agreement with Han et al. [24], for the halophyte *Kosteletzkya virginica*. In HM-stressed plants, glutathione has two functions: it is a primary antioxidant and a precursor of PCs involved in HM complexation and vacuolar sequestration. In the current research and regarding Cd + NaCl-untreated plants, the abundance of phytochelatin production may be explained by abundant glutathione, since glutathione is the substrate for phytochelatin biosynthesis. In the case of combined Cd and NaCl treatments, however, this theory does not match the present situation. NaCl induced an increase in Cd accumulation in the shoots, though this did not lead to higher PCs content in Cd + NaCl-treated plants; moreover, PCs content was even lower than in plants exposed to heavy metals in the absence of NaCl, in addition to the nontoxic appearance due to Cd deposition that occurred (Table 2), suggesting that other strategies may be adopted, herein, by the plant to cope with a higher Cd content. The first proposed strategy was demonstrated by Lutts and Lefèvre [37]: although the ability of synthesis PCs and the presence of functional PCs are ancestral characters to cope with high HMs doses, HM-tolerant plants rarely use this expensive strategy to detoxify HMs, as they instead overproduce organic acids that are efficiently involved in the chelation and sequestration of metals to non-active compartments. Another suggested mechanism that may be likely integrated, herein, in avoiding toxic cellular Cd in *S. fruticosa* under combined Cd and salinity, is through binding with an inorganic anion, chloride. High NaCl concentration increases the proportion of Cd fixed to the mineral fraction (chlorides). Complexation of Cd with Cl results in Cd-Cl formation. Cd-Cl is widely known in halophytes under combined Cd and salinity [38]. Hence, *S. fruticosa* with high-salt-concentration treatment had higher chloride concentration than low-salt-treated plants. Thus, a reduction in PCs content was more for high-salinized plants as, herein, the chloride-binding mechanism is the substitute for chelating by organic-compound PCs. On the other hand, Hamed et al. [28] and Ghnaya et al. [15] suggested that NaCl treatment could eliminate the most toxic form of Cd (Cd^2+^) in favor of another form bound to chloride anions. Accordingly, all previous findings suggest that the enhanced binding of Cd chloride anions induced by NaCl could play an important role in the amelioration of Cd tolerance by salinity and is also considered as a protective strategy against Cd in *S. fruticosa*. More in-depth research on this aspect is, therefore, required to be asserted.

### 3.6. Enzymatic Antioxidant Capacities as Affected by Cd Stress and Salinity Co-Occurrence in S. fruticosa

Under different stresses, plants produce various scavenging enzymes such as SOD, APX, and GPX, which play an essential role in modulating overproduction of ROS and keeping cellular homeostasis. Moreover, displaying non-noticeable responses in these scavenging enzymes along the experiment period and for up to 200 mM NaCl, compared with control NaCl-untreated plants, indicated no excess accumulation of ROS and that the plants were not suffering from oxidative stress. For the halophytes, it was reported that the concentration of NaCl, which first causes a significant oxidative injury and increased lipid peroxidation, thus inducing antioxidant activity, was 400 mM in *Salicornia brachiate* species and *Suaeda salsa* after 14 and 7 days of salinity exposure, respectively [39]. Accordingly, this study’s salt concentration (200 mM) was relatively low, to trigger severe oxidative stress for this halophytic species. The ROS generated from HM stress were trapped with the co-function of antioxidant enzymes. Therefore, *S. fruticosa* induced the enhancement of their activities. Salt co-occurrence imposes further elevated levels of ROS scavenger enzymes activities, SOD, APX, and GPX, and their substrates (parallel to the increase in their substrates ASA and GSH), revealing a powerful antioxidant system that trapped the exacerbation of the toxic ROS. The high--salt-treated plants (100–200 mM) had much more enzyme activity and less Cd toxicity than the low-salt-treated ones (50 mM), which correlated with enhanced plant growth. PO in the present study showed heterogeneous activities, whereas PO (involved in lignin biosynthesis in the plant cells) activity was activated in response to Cd stress because it prevents the entrance of HMs in plants via lignin production, which acts as a mechanical barrier [3]. Lignification in the cell wall induced by PO activity may involve the destruction of the photosynthetic apparatus due to aging and senescence, revealing restriction of the growth of stressed Cd-impacted cells, which results in aged leaves. The healthy cells displayed adequate lignin content and lessened PO activity, which may occur in the case of 0-Cd + NaCl-treated plant leaves, implying that salinity may ameliorate this effect by sufficiently reducing PO activity in Cd-affected plants. Zhou et al. [40] reported that simultaneously imposed Cd + NaCl improved the plants’ all-senescence-related parameters, as indicated by a significant decrease in PO. The marked increase in PO in the presence of HMs may be a valuable indicator of soils polluted with HM, which is parallel with the results of Nimptsch et al. [41], who recommended this enzyme as a feasible perspective biomarker for identifying HMs contamination.

## 4. Materials and Methods

### 4.1. Plant Material Collection

*S. fruticosa* dried stems, bearing fruits (the fruit corresponds to an utricule), were collected from Miami Island, Alexandria (31°16′04″ N 29°59′43″ E), from the coastline’s salt marshes along the Mediterranean in winter 2021. Seeds were extracted from dried fruits, before being preserved to be germinated during summer 2022 in pots filled with collected soil from *Salicornia*’s natural habitat and irrigated on alternate days for experimental purposes.

### 4.2. Hydroponic Culturing

The hydroponic medium was utilized to minimize the soil’s confounding factors, including changes in soil Cd chemistry, soil pH, and soil water potential caused by salt addition. It was prepared using ¼ Hoagland’s [42] and supplemented with 0, 50, 100, or 200 mM NaCl. According to Marco et al. [23], the plant exhibits a great tendency to show optimal growth at 200 mM NaCl, but at 300 mM, it was noticeably decreased, so the applied NaCl doses, herein, never exceed this limit (200 mM). The 2-week-old healthy plants were grown in the nutrient solution for one week for acclimation under 20–30 °C temperature, 16 h light and 8 h dark photoperiodic cycle at room temperature, with 50% relative humidity as well as light intensity of 350 μmol m^−2^ s^−1^. Four sets of five plants with identical size, fresh weight (0.9~1.2 g per plant), and similar health conditions were grown in 250 mL nutrient solution for each treatment. Stock solution of Cd was added to obtain the final concentrations of 25 μg L^−1^ CdCl_2_.H_2_O (L-Cd) and 50 μg L^−1^ CdCl_2_.H_2_O (H-Cd) in the nutrient solutions. Cd concentrations were chosen to proportionally mimic the actual Cd concentrations along the coastline of the Mediterranean Sea; Aquatic Ecosystems of Egypt according to El-Sorogy et al. [8]. The final Cd concentrations of the nutritive solutions were tested by using atomic absorption spectrophotometer (AAS, PerkinElmer A Analyst 200) has a detection limit of 0.001 mg L^−1^. The pH of the nutrient solutions was measured during experimentation period by pH meter and adjusted by HCl/NaOH, if required, to maintain 6.9. Cd concentrations were added as a single dose. All flasks were sealed with a preservative film and bumf at the plant crown to prevent water evaporation or chemical escape and wrapped with aluminum foil to prevent algal growth. Therefore, the growth media did not need to be replenished or changed during treatment duration (10 days). The Cd-untreated plants received 0 μg L^−1^ CdCl_2_.H_2_O was mentioned as 0-Cd. Plants grown in nutritive solutions without adding NaCl or Cd were used as controls and mentioned as (0 mM NaCl, 0-Cd). After 10 days of treatment, plants were collected, and the roots were submerged for 15 min in 25 mm EDTA-Na_2_ solution to remove Cd from the root surface. Four plants were randomly chosen from each treatment to estimate the following criteria.

### 4.3. Phenotypic Criteria

Root length was recorded and expressed in cm. Afterward, plants were divided into roots (belowground) as well as shoots including stems and leaves (aboveground). The samples of shoots were randomly divided into two groups; one was used to evaluate the fresh and dry biomass of the aboveground, which was immediately dried at 70 °C in an oven to constant weight. The other group was used to collect green leaves, which were rinsed with sterile distilled water and frozen in liquid nitrogen before being stored at −80 °C for further analyses.

### 4.4. Shoot Succulence Degree (SSD)

SSD was calculated by measuring shoot fresh weight and its dry weight. Given values are means from 4 individual samples and four replicate experiments per treatment factor [43].
SSD (g g^−1^) = Shoot fresh weight/Shoot dry weight 

### 4.5. Transpiration Rate

The transpiration rate was assessed using the method of Llanes et al. [44]. Transpiration was evaluated indirectly by recording alterations in the volume of a culture solution, with the same conditions as those described for the hydroponic cultures. One plant per treatment was placed in the holed rubber stopper of a transparent graduated cylinder having a definite volume of solution, which was then sealed with silicone for 24 h. The amount of solution consumed was determined after 24 h, under the same photoperiod circumstances as the hydroponic culture to assess the volume of transpired water. The mL of transpired water per leaf weight was estimated, and the result was reported as mL transpired H_2_O g^−1^ FW day^−1^.

### 4.6. Total Osmotic Potential (TOP) Determination

TOP was estimated. The leaf sap was prepared, using the technique proposed by Abeed and Dawood [45], by crushing fresh leaves, followed by centrifugation for 15 min at 10,000× *g*, and the resulting extract was utilized to measure the osmotic potential utilizing TridentMed’s 800 CL Osmometer. The osmotic potentials (MPa) were then calculated using Walter’s tables [46].

### 4.7. Proline Determination

As illustrated by Bates et al. [47], free proline was assessed in dry leaves. Homogenization of leaf samples was done in 3% sulfosalicylic acid (6 mL), before centrifugation at 10,000× *g*. The supernatant (2 mL) was blended with glacial acetic acid (2 mL) as well as ninhydrin. Sample heating was done for one hour at 100 °C, before cooling to room temperature. Extraction of the reaction mixture was performed with toluene (4 mL), and the content of free toluene was estimated to be 520 nm, which was expressed as milligrams per gram (dry weight). A calibration curve was established using proline solutions ranging from 0.05 to 1.5 mM, in the same medium as the one used for the extraction, and the data were expressed as μmol g^−1^ FW.

### 4.8. Enzymatic and Non-Enzymatic Antioxidant Capacities

For non-enzymatic antioxidants such as ascorbic acid (ASA) and reduced glutathione (GSH), the supernatant of freshly ground leaves in trichloroacetic acid was used to quantify ASA and GSH, using procedures developed by Jagota and Dani [48] and Ellman [49], respectively. According to Nahar et al. [50], phytochelatins (PCs) were calculated by subtracting the quantity of GSH from non-protein thiols, which were produced by combining the supernatant of crushed leaves and sulfosalicylic acid with Ellman’s reaction mixture [49].

The homogenization of each treatment’s fresh leaves was done in a mortar and pestle with sodium phosphate buffer 0.05 M (pH 7.5). The centrifugation of homogenate was done for 20 min at 10,000 r/min, and the supernatant was used to analyze leaf enzymatic potential, as identified by scanning Glutathione peroxidase (GPX/EC.1.11.1.9, μmol mg^−1^ protein g^−1^ FW min^−1^), ascorbate peroxidase (APX; EC1.11.1.11, μmol mg^−1^ protein g^−1^ FW min^−1^), and (SOD/EC.1.15.1.1, μmol mg^−1^ protein g^−1^ FW min^−1^), by the methods of Flohé and Günzler [51], Abeed et al. [52], and Abeed et al. [53], respectively. The peroxidase activity (PO, U mg^−1^ protein min^−1^) was quantified following enzyme extraction from leaves, as described by Ghanati et al. [54]. The PO activity was assessed according to the absorbance increase at 470 nm, utilizing 168 mM guaiacol in H_2_O_2_ (30 mM) and phosphate buffer (100 mM). The absorbance change was altered to units (U), using 26.6 mM^−1^ cm^−1^ extinction coefficient.

### 4.9. Cation Assay

Grounding desiccated samples was performed until obtaining a fine powder with a pestle as well as porcelain mortar, before being digested in a 4:1 (*v*/*v*) solution of HNO_3_-HClO_4_. The K and Na content in the homogenate were measured following the flame emission method (Carl-Zeiss DR LANGE M7D flame photometer, Carl-Zeiss AG, Jena, Germany) [45]. Instrument detection limits in mg g^−1^ DW were 0.23 and 0.15 for Na and K, respectively, while the method quantitation limits in mg g^−1^ DW were 1.02 and 0.96 for Na and K, respectively.

An atomic absorption/flame emission spectrophotometer (Shimadzu-model AA-630-02, Shimadzu Corporation, Kyoto, Japan) was used to measure the amounts of Ca and Cd in the same homogenate. 

Quality assurance and quality control (QA/QC) for Cd internal standards (CPAchem, Bogomilovo, Bulgaria) were used while standardizing the equipment with certified reference material (SRM 1547, peach leaves), as adopted by Eissa and Abeed [2]. Given values are means from 4 individual samples and four replicate experiments per each treatment factor. The calibration curve was prepared from the cadmium atomic absorption standard solution of 1000 μg mL^−1^ Cd in 1% HNO_3_ (Sigma Aldrich, St. Louis, MO, USA). The peach leaves as a reference material (SRM 1547) were measured at the same time for quality assurance. The recovery rate was 90%–105%. Instrument detection limits in mg g^−1^ DW were 0.40 and 0.04 for Ca and Cd, respectively, and the method quantitation limits in mg g^−1^ DW were 2.2 and 0.55 and 0.52 for Ca and Cd roots and shoots, respectively.

### 4.10. Cd Accumulation Characteristics

The halophyte *S. fruticosa* phytoremediation potential was estimated according to [55,56], via calculation of the following indicators: Bioconcentration factor (BCF) and enrichment factor = Cd concentration in the plant/Cd concentration in external medium;Translocation factor (TF) = Cd concentration in the shoot/Cd concentration in the root;Cd absorption efficiency (AE) = Cd accumulation in the whole plant/root biomass.

### 4.11. Statistical Analysis

The obtained data were collected from four randomly chosen plants/replicates/treatments inserted into plot mean basis analysis and evaluated utilizing the 21st version of SPSS software. The one-way evaluation of variance was followed by a post hoc test (Tukey’s multiple range tests). The level of statistical significance was set at (*p* < 0.05).

## 5. Conclusions

Since *S. fruticosa* is a euhalophyte, with salinity-tolerance capabilities that may indirectly lead to Cd tolerance. Without Cd, the variations in most of the investigated parameters between low-salt-affected and high-salt-affected plants were insignificant, owing primarily to the beneficial osmotic potential of salt. Cd toxicity was substantially more severe in low-salt-treated plants than in high-salt-treated plants. Cd toxicity mechanisms in *S. fruticosa* include significant disruption of plant–water interactions as well as the activation of aging and senescence-mediated enzymes’ peroxidase (PO). *S. fruticosa* demonstrated an adequate transpiration rate and shoot succulent degree, which may aid in the maintenance of plant water content, as well as a large amount of aboveground biomass production and deep rooting and efficient management of oxidative stress via elevated levels of AsA + GSH and enzyme activity modulation. Furthermore, *S. fruticosa* sequesters heavy metals intercellularly, rather than in the vacuole, as shown by the decreased PCs by salinity, indicating a positive role in phytoextraction. The salt-induced increase in Cd tolerance refers to the possibility of utilizing *S. fruticosa* for Cd phytoextraction. In the current investigation, the efficacy of employing *S. fruticosa* to remediate and enhance Cd-contaminated saline media is restricted to the kind of heavy metal and the dosages used. More research should be focused on the effectiveness of *S. fruticosa* in removing additional heavy metals from saline soils and/or from other substrates presenting a high electrical conductivity.

## Figures and Tables

**Figure 1 plants-11-02556-f001:**
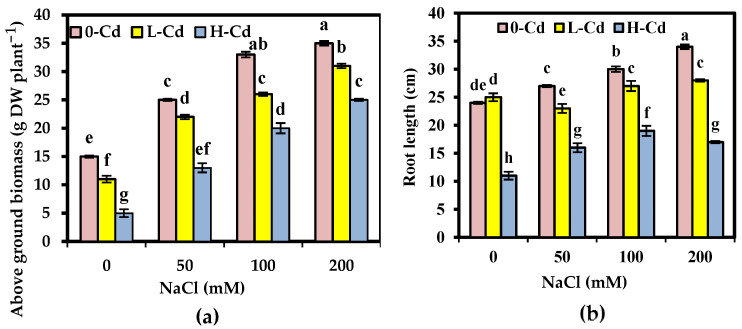
Aboveground biomass (g DW plant^−1^) and root length (cm) (**a**,**b**) of *Salicornia fruticosa* exposed to nutrient solution containing 0, 25, and 50 μg L^−1^ Cd (0-Cd, L-Cd, and H-Cd, respectively) without or with 50, 100, and 200 mM NaCl. Each value is the average of four replicates ± SE. Values bearing different letters are significantly different at *p* < 0.05 based on Tukey’s test. DW: dry weight.

**Figure 2 plants-11-02556-f002:**
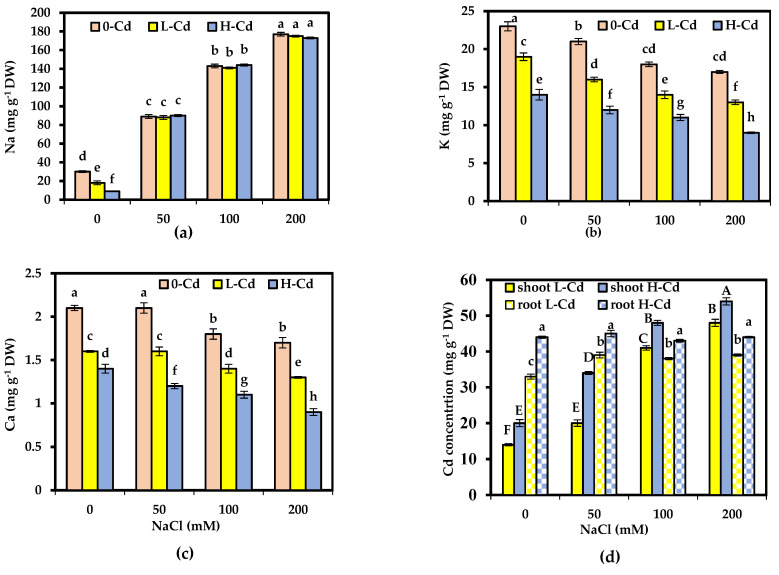
Concentrations in mg g^−1^ DW of leaf sodium (Na), potassium (K), calcium (Ca), and shoot and root cadmium (Cd) (**a**,**b**,**c**,**d**, respectively) in *Salicornia fruticosa* exposed to nutrient solution containing 0, 25, and 50 μg L^−1^ Cd (0-Cd, L_._-Cd, and H_._-Cd, respectively) without or with 50, 100, and 200 mM NaCl. Each value is the average of four replicates ± SE. For shoot and root Cd concentration, values of 0-Cd were removed from (**d**), as no Cd was detected in 0-Cd treatment. Values bearing different letters are significantly different at *p* < 0.05 based on Tukey’s test. DW: dry weight.

**Figure 3 plants-11-02556-f003:**
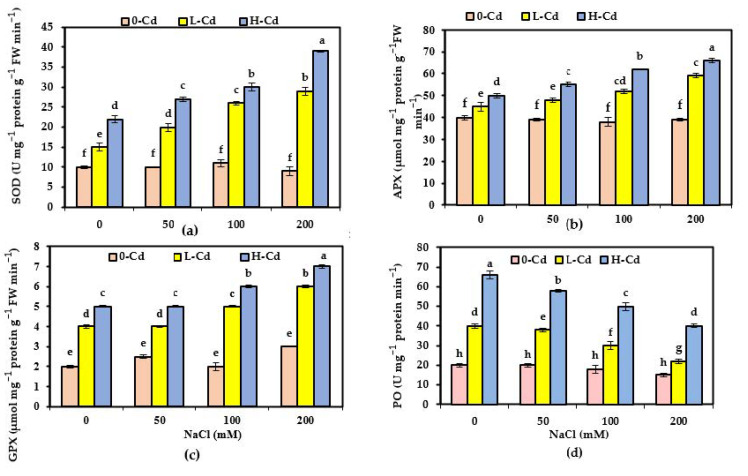
Activities of superoxide dismutase (SOD; U mg^−1^ protein g^−1^ FW min^−1^), glutathione peroxidase (GPX; μmol mg^−1^ protein g^−1^ FW min^−1^), ascorbate peroxidase (APX; μmol mg^−1^ protein g^−1^ FW min^−1^), and peroxidase (PO; U mg^−1^ protein min^−1^) (**a**,**b**,**c**,**d**, respectively) in leaves of *Salicornia fruticosa* exposed to nutrient solution containing 0, 25, and 50 μg L^−1^ Cd (0-Cd, L-Cd, and H-Cd, respectively) without or with 50, 100, and 200 mM NaCl. Each value is the average of four replicates ± SE. Values bearing different letters are significantly different at *p* < 0.05 based on Tukey’s test.

**Table 1 plants-11-02556-t001:** Transpiration rate (mL transpired H_2_O g^−1^ FW day^−1^), shoot succulence degree (g g^−1^), and total osmotic potential (TOP; MPa) in leaves of *S. fruticosa* exposed to nutrient solution containing 0, 25, and 50 μg L^−1^ Cd (0-Cd, L-Cd and H-Cd, respectively) without or with 50, 100, and 200 mM NaCl. Each value is the average of four replicates ± SE. Values bearing different letters are significantly different at *p* < 0.05 based on Tukey’s test.

Treatments		Transpiration Rate (mL Transpired H_2_O g^−1^ FW Day^−1^)	Shoot Succulence Degree (g g^−1^)	TOP (MPa)
0 mM NaCl	0-Cd	30 ± 0.50 ^a^	5.02 ± 0.04 ^a^	−24.7 ± 0.32 ^f^
	L-Cd	22 ± 0.40 ^b^	5.00 ± 0.05 ^a^	−25.5 ± 0.22 ^e^
	H-Cd	6 ± 0.08 ^d^	3.45 ± 0.06 ^c^	−28.3 ± 0.15 ^a^
50 mM NaCl	0-Cd	28 ± 0.60 ^a^	5.41 ± 0.04 ^a^	−24.7 ± 0.25 ^f^
	L-Cd	20 ± 0.90 ^b^	5.11 ± 0.03 ^a^	−25.1 ± 0.13 ^e^
	H-Cd	11 ± 0.40 ^c^	3.99 ± 0.05 ^b^	−27.8 ± 0.14 ^b^
100 mM NaCl	0-Cd	30 ± 0.90 ^a^	5.33 ± 0.04 ^a^	−24.9 ± 0.44 ^f^
	L-Cd	21 ± 0.60 ^b^	4.99 ± 0.03 ^a^	−24.8 ± 0.24 ^f^
	H-Cd	19 ± 0.70 ^b^	4.11 ± 0.03 ^b^	−26.4 ± 0.13 ^c^
200 mM NaCl	0-Cd	29 ± 0.70 ^a^	5.34 ± 0.02 ^a^	−24.8 ± 0.12 ^f^
	L-Cd	20 ± 0.60 ^b^	5.11 ± 0.02 ^a^	−24.7 ± 0.25 ^f^
	H-Cd	22 ± 0.60 ^b^	4.28 ± 0.01 ^b^	−25.8 ± 0.44 ^d^

0-Cd: no cadmium added; L-Cd: low cadmium concentration; H-Cd: high cadmium concentration; FW: fresh weight; g: gram; mL: milliliter; TOP: total osmotic potential; MPa: mega pascal.

**Table 2 plants-11-02556-t002:** Effect of NaCl on some Cd phytoremediation parameters in *Salicornia fruticosa*: accumulated cadmium root and shoot (μg plant^−1^ DW); bioaccumulation factor (BCF); translocation factor (TF); and absorption efficiency (AE; µg g^−1^). Values bearing different letters are significantly different at *p* < 0.05 based on Tukey’s test.

Treatments		Accumulated Cd (μg Plant^−1^ DW)		BCF	TF	AE (µg g^−1^)	Phytoremediation Strategy
		Root	Shoot				
0 mM NaCl	L-Cd	12.3 ± 0.10 ^c^	4.4 ± 0.2 ^h^	1.1± 0.07 ^f^	0.42 ± 0.01 ^f^	221 ± 3.5 ^g^	Phytostabilization
	H-Cd	15.0 ± 0.20 ^a^	6.1 ± 0.2 ^g^	1.4 ± 0.08 ^f^	0.45 ± 0.02 ^f^	404 ± 5.0 ^e^	
50 mM NaCl	L-Cd	11.0 ± 0.10 ^d^	8.9 ± 0.3 ^f^	5.3 ± 0.10 ^e^	0.51 ± 0.03 ^e^	340 ± 4.8 ^f^	Phytostabilization
	H-Cd	13.0 ± 0.20 ^b^	16.0 ± 0.4 ^e^	6.8 ± 0.20 ^d^	0.76 ± 0.05 ^d^	611 ± 3.2 ^c^	
100 mM NaCl	L-Cd	2.5 ± 0.05 ^e^	23.0 ± 0.5 ^d^	7.4 ± 0.10 ^c^	1.08 ± 0.01 ^c^	552 ± 3.0 ^d^	Phytoextraction
	H-Cd	2.9 ± 0.04 ^e^	43.0 ± 0.6 ^c^	9.9 ± 0.20 ^b^	1.12 ± 0.06 ^b^	907 ± 4.1 ^b^	
200 mM NaCl	L-Cd	1.4 ± 0.07 ^f^	55.0 ± 0.9 ^b^	11.3 ± 0.30 ^a^	1.23 ± 0.07 ^a^	690 ± 4.4 ^c^	Phytoextraction
	H-Cd	1.9 ± 0.05 ^f^	62.0 ± 0.8 ^a^	12.1 ± 0.30 ^a^	1.23 ± 0.07 ^a^	1303 ± 6.6 ^a^	

No Cd was detected in 0-Cd treatments, so they were not included in the table. L-Cd: low cadmium concentration; H-Cd: high cadmium concentration; DW: dry weight.

**Table 3 plants-11-02556-t003:** Concentrations of ascorbic acid (ASA; µmol g^−1^ FW), reduced glutathione (GSH; nmol g^−1^ FW), phytochelatins (PCs; µmol g^−1^ DW), and proline (μmol g^−1^ FW) in leaves of *Salicornia fruticosa* exposed to nutrient solution containing 0, 25, and 50 μg L^−1^ Cd (0-Cd, L.-Cd, and H.-Cd, respectively) without or with NaCl 50, 100, and 200 mM. Each value is the average of four replicates ± SE. Values bearing different letters are significantly different at *p* < 0.05 based on Tukey’s test.

Treatments		ASA (µmol g^−1^ FW)	GSH (nmol g^−1^ FW)	PCs (µmol g^–1^ DW)	Proline (μmol g^−1^ FW)
0 mM NaCl	0-Cd	2.60 ± 0.05 ^e^	110 ± 2.1 ^k^	10.91 ± 0.9 ^g^	5.2 ± 0.07 ^i^
	L-Cd	3.50 ± 0.04 ^d^	170 ± 1.5 ^h^	28.62 ± 0.4 ^f^	11.6 ± 0.41 ^e^
	H-Cd	1.40 ± 0.01 ^h^	201 ± 1.0 ^g^	37.33 ± 0.3 ^a^	21.0 ± 0.52 ^a^
50 mM NaCl	0-Cd	2.51 ± 0.01 ^e^	113 ± 2.0 ^j^	10.51 ± 0.5 ^g^	5.1 ± 0.01 ^i^
	L-Cd	3.90 ± 0.02 ^d^	210 ± 1.4 ^f^	27.03 ± 0.4 ^f^	9.8 ± 0.21 ^f^
	H-Cd	1.60 ± 0.01 ^h^	260 ± 2.3 ^e^	35.11 ± 0.2 ^b^	19.3 ± 0.30 ^b^
100 mM NaCl	0-Cd	2.61 ± 0.03 ^e^	116 ± 1.1 ^i^	11.01 ± 0.5 ^g^	4.9 ± 0.01 ^i^
	L-Cd	4.51 ± 0.03 ^b^	269 ± 2.3 ^d^	22.24 ± 0.4 ^e^	8.5 ± 0.11 ^g^
	H-Cd	2.01 ± 0.01 ^f^	304 ± 3.0 ^b^	31.81 ± 0.8 ^c^	16.6 ± 0.20 ^c^
200 mM NaCl	0-Cd	2.54 ± 0.01 ^e^	118 ±1.1 ^i^	10.70 ± 0.2 ^g^	5.0 ± 0.09 ^i^
	L-Cd	5.12 ± 0.03 ^a^	297 ± 1.6 ^c^	20.54 ± 0.2 ^d^	7.2 ± 0.07 ^h^
	H-Cd	2.55 ± 0.01 ^e^	366 ± 2.2 ^a^	30.25 ± 0.7 ^c^	15.4 ± 0.13 ^d^

Cadmium added. L-Cd: low cadmium concentration; H-Cd: high cadmium concentration; FW: fresh weight; DW: dry weight; g: gram.

## Data Availability

Not applicable.

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
