# Peer review of "Cd Phytoextraction Potential in Halophyte Salicornia fruticosa: Salinity Impact"

_plants, 2022, doi:10.3390/plants11192556_

Round 1
Reviewer 1 Report (Previous Reviewer 2)
The MS has been improved significantly and I am happy with the corrected version to be published.
Author Response
Comment :The MS has been improved significantly and I am happy with the corrected version to be published.
Response: We thankfully acknowledge your appreciation and positive comments on our manuscript. We are very glad that the revised version of our Ms is endorsed by you.

Reviewer 2 Report (Previous Reviewer 1)
The manuscript has been improved, but is still not in a publishable form.
Abstract – overly long; has in fact been lengthened by several sentences of results which is good, but the background info needs to be edited out
Introduction – there has been a general trend over the last years to replace the term ‘heavy metals’ with ‘potentially toxic elements’ (e.g. International Journal of Environmental Research and Public Health, MDPI, 2019, 16 (22), pp.4446. ff10.3390/ijerph16224446ff. ffhal-02889766)
- - Still quite long, but better
Materials and Methods – this section should follow the Introduction and precede the Results
Results – In the case of all the figures, there continues to be the issue of starting with the wrong letters (letters indicating statistical significance should start at a in all cases; ditto for all the tables)
Discussion – I find this section much more readable now with the subheadings!
- However, I still find it overly long and repetitive of the results section in some cases
Materials and Methods – I have never seen a journal in which this section follows after the Discussion; it belongs immediately following the Introduction
- - QA/QC section is still missing
Author Response
Reviewer 2
“The manuscript has been improved, but is still not in a publishable form”
We appreciate your comments regarding the improvement of Ms. We greatly appreciate your time and consideration, and we hope that you would find our revised manuscript suitable now for publication
Comment 1: Abstract – overly long; has in fact been lengthened by several sentences of results which is good, but the background info needs to be edited out.
Rsponse: The background info was reduced to half (from 65 words to 33 words). Changes can be kindly tracked in abstract and highlighted in red.
Comment 2: Introduction – there has been a general trend over the last years to replace the term ‘heavy metals’ with ‘potentially toxic elements’ (e.g. International Journal of Environmental Research and Public Health, MDPI, 2019, 16 (22), pp.4446. ff10.3390/ijerph16224446ff. ffhal-02889766)
- Still quite long, but better
Response: We replace the term ‘heavy metals’ with ‘potentially toxic elements in many places in itroduction and use this paper [6] for documentation. Changes were highlighted in red.
We further shortened intorduction. Many sentences were removed. Changes can be kindly tracked in introduction in lines 52, 64, 75, 84, 93, 97.
Comment 3: Materials and Methods – this section should follow the Introduction and precede the Results
Materials and Methods – I have never seen a journal in which this section follows after the Discussion; it belongs immediately following the Introduction
Response: We are worried about such modification as we adopted the “author instructions” of Plants journal in which “Materials and Methods” is the fourth section in the Ms and comes after Discussion. Below, There are screen shots from Plants jouranl home page showing the instructions regarding research articles preparation and some relevant articles.
Zhou, L.; Cai, Y.; Yang, L.; Zou, Z.; Zhu, J.; Zhang, Y. Comparative Metabolomics Analysis of Stigmas and Petals in Chinese Saffron (Crocus sativus) by Widely Targeted Metabolomics. Plants 2022, 11, 2427. https://doi.org/10.3390/plants11182427
Runno-Paurson, E.; Nassar, H.; Tähtjärv, T.; Eremeev, V.; Hansen, M.; Niinemets, Ü. High Temporal Variability in Late Blight Pathogen Diversity, Virulence, and Fungicide Resistance in Potato Breeding Fields: Results from a Long-Term Monitoring Study. Plants 2022, 11, 2426. https://doi.org/10.3390/plants11182426
Xiong, Q.; Che, Y.; Dai, Q.; Liu, B.; Liu, G. Morphology and molecular phylogeny of Genus Oedogonium (Oedogoniales, Chlorophyta) from China. Plants 2022, 11, 2422. https://doi.org/10.3390/plants11182422
Zhang, R.; Li, H.; Gui, Y.; Wei, J.; Zhu, K.; Zhou, H.; Lakshmanan, P.; Mao, L.; Lu, M.; Liu, J.; Que, Y.; Li, S.; Liu, X. Comparative Transcriptome Analysis of Two Sugarcane Cultivars in Response to Paclobutrazol Treatment. Plants 2022, 11, 2417. https://doi.org/10.3390/plants11182417
Chen, J.; Li, J.; Ma, M.; Li, B.; Zhou, Y.; Pan, Y.; Fan, Y.; Yi, B.; Tu, J. Improvement of Resistance to Clubroot Disease in the Ogura CMS Restorer Line R2163 of Brassica napus. Plants 2022, 11, 2413. https://doi.org/10.3390/plants11182413
Comment 4: Results – In the case of all the figures, there continues to be the issue of starting with the wrong letters (letters indicating statistical significance should start at a in all cases; ditto for all the tables)
Response: In the first version smallest value bear letter of significance a bigger bear b, c, d and so on,…….. . But, in the second version we have corrected this as reviewer recommendation; the biggest value bear letter of significance a, smaller bear b then c,d and so on. Thus letters indicating statistical significance start at a, then b, then c then d and so on in all current represented figures and tables.
Comment 5: Discussion – I find this section much more readable now with the subheadings!
However, I still find it overly long and repetitive of the results section in some cases
Response: Some repetitions of the results section in lines 272, 285, 361, 390 were removed. Changes can be kindly tracked in Discussion section.
Comment 6: - QA/QC section is still missing
Response: QA/QC was added. Changes were highlighted in red and can be kindly tracked in line 553

This manuscript is a resubmission of an earlier submission. The following is a list of the peer review reports and author responses from that submission.
Round 1
Reviewer 1 Report
This manuscript contains some interesting research on the effects of NaCl on Cd accumulation in Salicornia fructicosa, but I do not recommend publication and note that the manuscript would require extensive re-working prior to being considered again. I provide some specific comments which I hope will be useful to the authors.
Introduction
- Overly long; focus only on published Salicornia research and the intentions of the research undertaken
- Several nonsensical (e.g. grammatically incorrect) sentences make interpretation difficult for the reader (e.g. lines 77-78, lines 96-97, lines 106-108, etc.)
Materials and Methods
- This information is currently in section 4 and needs to be moved to section 2, immediately following the Introduction
- Background info on S. fructicosa belongs here
- Some explanation on what is meant by ‘dried plants’
- Entirely unclear why hydroponic culturing was method employed when the authors repeatedly talk about ‘soil contamination’; no attempt made to explain how/why hydroponics are being used and how this research might (or might not) translate to soil systems
- Unclear exactly how many plants were grown (n=10 for each treatment?) and how the four replicates referred to in the figures were selected from amongst these
- Unclear if leaves were considered as part of ‘upground’ (change to above ground please) parts of the plant or removed first for other analyses
- Transpiration rate ‘assessment’ very unclear as the authors conclude that section with ‘transpired water per leaf weight was estimated’ – no idea what this means
- No explanation of replicates or controls in any of the methods
- Where is the section detailing quality assurance/quality control; - i.e. what controls were used? what detection limits were determined? were certified standards were employed? How many internal duplicates were run? Etc.
- Some of this info could be included in the statistical analysis section which is lacking many details
Results
- First and foremost, this section is overly long; - text should briefly describe trends only without interpretation (e.g. statements such as in lines 114-116 ‘These observations demonstrated….’ do not belong here, but rather in the Discussion section
- Figure 1 – why was the above ground biomass measured, but the root length; - one would usually go with biomass of both or length of both; letters indicating statistical significance should start at a (not c) or in the case of 1b, should not start with b (but rather a)
- Tables 1,2 and 3 – without proper methods it is not possible to assess whether the significant digits (which seem to vary) are appropriate
- Figures 2 and 3 – some of the same issues as in Figure 1
Discussion
- Currently 4.5 pages single spaced with no breaks; - this is overly long and wordy (reads more like a literature review) and needs to be made much more concise referring specifically to your own data only and how it relates to the literature
- Use of subheadings to correspond to your results would make this section far more readable
Conclusions
- Again, shorten as this section will immediately follow the Discussion once the manuscript is properly arranged
Reviewer 2 Report
Overall, the study provides valuable information about the effect of salinity on Cd uptake, accumulation, and translocation in the halophyte species S. fruticose, and suggests the possibility of using this species for Cd phytoextraction in saline soils polluted with heavy metals. The MS is well-written and interesting to read. However, there are some areas in the MS that deserve improvements. I would recommend this MS for publication with suggested clarifications/corrections.
1. The plants were treated with 25 µg/L and 50 µg/L of Cadmium (Cd), and the authors assigned them as low and high concentrations. However, there is no information on why those two concentrations were selected for this study. Are they environmental relevant? Please provide relevant information in the introduction and section 4.2.
2. It is not clear whether Cd concentrations were maintained during the experiment (e.g, changing media solution daily) or just one-off treatment. It is also important to confirm those two Cd concentrations were actually in the treatment solutions (e.g., using AAS instrument to check the actual concentrations
3. In the cation assay section, information about detection and quantification limits of the instruments for measured elements (Ca, K, Na and Cd) needs to be provided.
4. Typo errors: Please check typo errors throughout the MS. Some examples are as below:
- Salicornia fruticose was first abbreviated as S. fruticose in the introduction, line 96. However, the full term is seen in the result and discussion sections again (line 113, 119, 122, 241, 246, 260, 261). Please correct them.
- In all figure captions, authors used lower case letters (e.g, a, b, c…), however, in the main text, upper case letters (A, B, C…) were used. Please correct them.
- Line 143: "Table 1", not "Tab 1".
- Upper case “P” and lower case “p” were inconsistently used in the MS (line 132, 153, 173, 190, 219, 235, 568). The p should be lowercase and italicized. Please correct them.
Reviewer 3 Report
The authors Salama et al. studied the effects on Salicornia fruticose of cadmium, under different salinity conditions. The study analyzes compartmentalization and concentration variation of various parameters and variables in relation to the variation of salt and Cd in the culture medium: Cd, minerals, phytochelatins, enzymes and antioxidant compounds.
The data collected is very interesting and worthy of publication, but unfortunately, in my opinion, the article is overall not suitable for publication. Although I am not qualified for an in-depth assessment of the English language, I have found considerable difficulty in reading this text. The language used is often imprecise and sometimes uses improper terms. Many sentences introduce concepts that are repeted many times throughout the manuscript. It would be necessary adopt a style synthetic and clear, instead than propose some general information (for instance concerning the ability of this plant to tolerate Cadmium or similar concepts) and then taken up again the same with minimal variations in different points of the text. This redundancy makes it difficult for the reader to follow a logical thread. To better understand this generic comment, please consider my notes in the pdf that I am attaching. By way of example, concerning the need of synthesis, please consider the following text (lines 76-84) in your version (108 words) and in a revised form that I find clearer and shortened by 20% (86 words):
“Additionally, multiple investigations have shown that some halophytes may be resistant to HMs and accumulate significant bioavailable HMs concentrations in their tissues [7]. Because of processes that impart tolerance to ions other than chloride and sodium. Evidence shows that halophytes evolutionary adaptations may potentially offer resistance to other harmful substances [10]. Therefore, halophytes are the optimum species of plants for remediating HM-contaminated salty soils. Halophytes high tolerance of metals substantially correlates with salt tolerance traits such as antioxidant systems [11]. Osmoprotectant production includes proline to scavenge free radicals as well as retrain the balance of water [11, 12] and salt gland excretion onto the surface of the leaf” (108 words)
The same concepts could be expressed as follow:
“Additionally, multiple investigations have shown that some halophytes may be resistant to HMs and accumulate significant bioavailable HMs concentrations in their tissues [7]. Indeed, halophytes have developed adaptive mechanisms against saline stress based on osmoprotectants, which includes proline [11, 12], and the ability to excrete salt onto the leaf surface by means of specialized glands. Interestingly, the heavy metal tolerance of halophytes appears to be related to their salt tolerance and antioxidant systems [11] and this makes halophytes suitable to exploited for remediating HM-contaminated salty soils” (86 words)
This lack of synthesis also determines a conceptual dispersion that impacts the understanding of the text. In addition, some terms are sometimes incorrectly used, in my opinion. For example, the abbreviation "HM" is frequently used in the text where "Cd" should be used. In fact, the data collected in relation to the treatment with Cd do not allow to believe that they can be extended to other heavy metals, even if this could be true in many cases.
The discussion is particularly burdened by the problems mentioned above. It is too long and dispersive and could be reduced by half. The data analysis should proceed by focusing on the problems related to the showed results, interpreting them for the strongly related cause-effect connections, while often unsustainable interpretations are inserted on the basis of references to other works. For example, please see my notes on page 9 of 17.
Some data show inconsistencies, or apparent inconsistencies, that are not discussed and cleared. For instance, in Fig. 2D the Cd concentration (mg g-1 plant DW) are almost stable in roots at different NaCl concentrations, but Table 2 show that the accumulated Cd (μg plant−1 DW) decreases in roots 5 folds or more. It is unclear how the accumulated metal can vary in different root-groups, if the concentration is stable and the root length is negatively affected by Cd (see fig. 1B). The opposite should happen.
I have found several inconsistencies that you can find highlighted in the first part of this long discussion, then, in the second part, I stopped marking them because they were too expensive in terms of commitment. However, the whole discussion should be reworked more critically and carefully.
Minor remarks.
- In the figures, the graphs on the right show a x-axis with an inverted scale. This is confusing and unusual. It is strongly recommended to maintain the usual orientation as in the graphs on the left.
-The abbreviation (FW, DW etc.) should be reported in full in the text at their first use.
